# An Innovative Fingerprint Location Algorithm for Indoor Positioning Based on Array Pseudolite

**DOI:** 10.3390/s19204420

**Published:** 2019-10-12

**Authors:** Lu Huang, Xingli Gan, Baoguo Yu, Heng Zhang, Shuang Li, Jianqiang Cheng, Xiaohu Liang, Boyuan Wang

**Affiliations:** 1State Key Laboratory of Satellite Navigation System and Equipment Technology, Shijiazhuang 050081, China; 18642720668@163.com (L.H.); yubg@sina.cn (B.Y.); 13582161539@163.com (H.Z.); lishuangcetc54@163.com (S.L.) a243945274@163.com (J.C.); liangxiaohu@126.com (X.L.); boyuan@hrbeu.edu.cn (B.W.); 2The 54th Research Institute of China Electronics Technology Group Corporation, Shijiazhuang 050081, China

**Keywords:** indoor localization, array pseudolite, carrier phase difference, fingerprint matching, deep neural network

## Abstract

Since the signals of the global navigation satellite system (GNSS) are blocked by buildings, accurate positioning cannot be achieved in an indoor environment. Pseudolite can simulate similar outdoor satellite signals and can be used as a stable and reliable positioning signal source in indoor environments. Therefore, it has been proposed as a good substitute and has become a research hotspot in the field of indoor positioning. There are still some problems in the pseudolite positioning field, such as: Integer ambiguity of carrier phase, initial position determination, and low signal coverage. To avoid the limitation of these factors, an indoor positioning system based on fingerprint database matching of homologous array pseudolite is proposed in this paper, which can achieve higher positioning accuracy. The realization of this positioning system mainly includes the offline phase and the online phase. In the offline phase, the carrier phase data in the indoor environment is first collected, and a fingerprint database is established. Then a variational auto-encoding (VAE) network with location information is used to learn the probability distribution characteristics of the carrier phase difference of pseudolite in the latent space to realize feature clustering. Finally, the deep neural network is constructed by using the hidden features learned to further study the mapping relationship between different carrier phases of pseudolite and different indoor locations. In the online phase, the trained model and real-time carrier phases of pseudolite are used to predict the location of the positioning terminal. In this paper, by a large number of experiments, the performance of the pseudolite positioning system is evaluated under dynamic and static conditions. The effectiveness of the algorithm is evaluated by the comparison experiments, the experimental results show that the average positioning accuracy of the positioning system in a real indoor scene is 0.39 m, and the 95% positioning error is less than 0.85 m, which outperforms the traditional fingerprint positioning algorithms.

## 1. Introduction

With the development of global navigation satellite systems (GNSS), people’s demand for positioning services is also increasing. The availability of location-based services depends largely on the accuracy of the location. Although global positioning systems (GPS) can provide high-precision positioning in an open environment, due to signal occlusion, GPS cannot work in complex indoor scenes [1]. Since the basic framework of indoor positioning was proposed in [2,3,4,5], more and more attention has been paid to the research of indoor positioning systems at home and abroad. At present, indoor positioning technology mainly includes: (1) Wi-Fi fingerprint positioning, mainly using real-time received signal strength and pre-built fingerprint map matching to achieve positioning, with a positioning accuracy of 5–10 m [6,7]. (2) Radio Frequency Identification (RFID), where the feature information of the target RFID tag (such as identity ID, received signal strength, etc.) is read by a fixed set of readers and the positioning is usually performed by the neighboring method, with a positioning accuracy of 1–3 m [8]. (3) Sound wave, most of the current use of reflective ranging method. The system consists of a main range finder and several electronic tags. The main range finder can be placed on the mobile robot body. Each electronic tag is placed in a fixed position in the indoor space [9]. (4) Visible light, which uses the intensity of light signal arrival or the flicker frequency of the light signal to achieve indoor positioning [10]. (5) Ultra-Wide Band (UWB) uses the anchor nodes and bridge nodes of known positions arranged in advance to communicate with the newly added blind nodes, and uses triangle positioning or fingerprint positioning to determine the position [11]. (6) Bluetooth Low Power (BLE), which usually adopts the proximity positioning method and fingerprint positioning method. Its coverage and positioning accuracy are related to the number of Bluetooth modules deployed, and the general positioning accuracy is 3–5 m [12,13]. (7) Infrared, where there are two specific implementation methods. One is to attach an electronic tag that emits infrared rays to the positioning object, and measure the distance or angle of the signal source through multiple infrared sensors placed indoors to calculate the position of the object. Another method is an infrared woven mesh, which directly locates the moving target by covering the space to be tested by an infrared ray woven by the transmitter and the receiver [14]. (8) Motion sensor-based inertial navigation, which mainly uses the motion data collected by inertial sensors, such as acceleration sensors, gyroscopes, and other information to measure the speed and direction of objects, and obtains the position information of objects based on pedestrian dead reckoning method [15,16], etc.

In this paper, an indoor positioning scheme based on pseudolite is proposed. Pseudolite is a ground-based transmitter that can transmit signals similar to global navigation satellite systems. Deployment of pseudolite in indoor environments can provide reliable positioning signals. Similarly, without changing the existing hardware of smartphones, it can provide the possibility of seamless switching indoors and outdoors. Pseudolite indoor positioning is actually realized by a pseudolite antenna, which transmits a similar-realistic satellite signal output by the navigation simulator to the user terminal, and uses these raw observation data to realize the indoor position prediction. In [17], a hyperbolic positioning method considering the geometric relationship between pseudolite antenna and receiver is introduced. The user position is solved by the least squares algorithm on the premise of the known initial position. In [18], an indoor positioning method combining Doppler positioning and carrier phase-based hyperbolic positioning with multi-channel GPS pseudolite is proposed. This method solves the problem of carrier phase ambiguity by modeling. In turn, the distance between the pseudolite and the receiver is obtained to achieve indoor positioning. In [19], a system that uses inertial navigation information to assist pseudolite positioning is introduced. The relative information provided by the inertial navigation system (INS) is used to obtain the ambiguity of the carrier phase to obtain relatively accurate distance information for the prediction of the receiver position. Since the traditional pseudolite data acquisition method is computationally intensive, and the acquisition method based on single-pulse detection cannot eliminate the influence of short pulses, this paper draws on the acquisition method proposed in [20] by utilizing the feature that the coherent integration result of a Direct Sequence Spread Spectrum (DSSS) pulse is robust to carrier Doppler estimation error, the three dimensional acquisition process is simplified to three one-dimensional searches. However, there are some challenging problems in scientific research and system design, such as indoor pseudolite and outdoor satellites have very different geometric properties. When positioning, it is usually necessary to obtain the initial position and solve the problem of carrier phase ambiguity. Moreover, the signal loss due to factors such as the near–far effect [16] or signal obstruction can have a great influence on the positioning result.

With the improvement of computer hardware level and the improvement of the deep learning model, the use of a multi-layer neural network to solve complex scientific problems has become the mainstream means. In this paper, the problem of indoor location prediction is transformed into one that can be solved by using a depth neural network model. Feature clustering is realized by feature extraction under latent space, which avoids training the model directly with original observation data, and reduces the influence of signal fluctuation caused by occlusion and multipath. Scholars in related fields have proposed some solutions. For example, in [21], a four-layer Deep Neural Network (DNN) network is proposed to generate a rough location estimate, and then a precise positioning model based on a hidden Markov model (HMM) is used to generate the final the location estimate. In [22], the application of two different types of restricted Boltzmann machines (RBMs) in indoor positioning is introduced, and the ray tracing technology is used to simulate and analyze different types of mobile wireless networks to verify the performance of the algorithm. In [23], the author introduces a DNN network composed of a stacked auto-encoder (SAE) and a feed-forward multi-level classifier for building and floor classification. It also provides some ideas for this paper. The paper [24] proposed a new DNN architecture consisting of an SAE for the reduction of feature space dimension. The paper [25] used an RBM to pre-train a deep neural network that takes as input the channel state information (CSI).

In this paper, a new neural network model is proposed. A variational auto-encoder (VAE) with position information constraints is used to acquire the distribution characteristics of pseudolite carrier phase fingerprint data in the latent space, which provides more representative feature knowledge for the positioning model. Compared with the model of [23], the proposed algorithm model has greater advantages in the model itself. In the coding stage, the input data is coded as a mixture of Gauss distribution rather than a single deterministic mapping, which improves the diversity of latent space, enables the input data to be reconstructed with the minimum error in the decoder stage, and makes the features obtained by the coding more representative. In model training, the location label is introduced as constraint information, which makes different locations have unique distribution and improves positioning performance. The main contributions of our work are as follows: We design an indoor positioning system based on multi-antenna array pseudolite. The system transmits a unique C/A code signal compatible with GPS/BDS (BeiDou Navigation Satellite System) satellite signal. Data acquisition and location are realized by using smart phones. The acquisition of the original observation data of pseudolite and the prediction of indoor position are realized by using smart phones.We propose a feature clustering method based on VAE in low-dimensional latent space. Firstly, in the stage of building the encoder model, four convolution layers are used to construct the hidden layer. Then, in the model learning stage, indoor location information is added as a prior condition to the training of the VAE, so that the model can learn the unique probability distribution at different indoor locations in low-dimensional latent space. Finally, the features are used to realize indoor location prediction. In this paper, we validate the clustering performance of the proposed model in different data sets.In this paper, pseudolite base station is built in the ordinary indoor office environment. The experimenters use the learning model to realize indoor positioning, and test the positioning accuracy under static and dynamic conditions, respectively. At the same time, in order to further compare the performance of the positioning system, we compare it with the commonly used fingerprint-based positioning methods. The experimental results show that the indoor positioning system can achieve the best positioning effect and meet the positioning requirements.

The remainder of this paper is organized as follows: Section 2 introduces the structure of the positioning system, and analyzes the realization principle and related preparations of the system. Section 3 introduces the structure of the innovative positioning model and the details of the model implementation algorithm, and verifies the clustering performance of the model on different data sets. The experimental area is described first, and results and performance evaluation are then reported in Section 4. Section 5 concludes the paper.

## 2. Preliminaries and Overview of System

Firstly, the concept of an array pseudolite system and carrier phase difference is introduced. Then, the time-varying characteristics of carrier phase difference and the spatial resolution of the signal in indoor environments are analyzed through measured data. Finally, the system framework and implementation details presented in this paper are introduced. At the same time, the principle of indoor fingerprint matching location based on carrier phase difference is elaborated.

### 2.1. Array Pseudolite System

Pseudolite’s signal transmitting baseband unit is a multi-channel radio frequency module (AD9371) driven by a digital signal processor (DSP) and field programmable gate array (FPGA). Each transmission channel is modulated into different C/A codes and navigation messages, and the signal frequency can be compatible with GPS L1 and Beidou B1. Since all channels of array pseudolites transmit signals at the same time (the same 1PPS), the time error of pseudolites received by the GNSS user receiver is the same, which can eliminate the influence of pseudolite clock deviation by data processing, Pseudolite base station and antenna are shown in Figure 1.

### 2.2. Carrier Phase Difference

We know that the pseudorange measurement information output by the pseudolite signal simulator is inaccurate, and the positioning method based on pseudorange cannot be used indoors. However, the carrier phase difference between each channel of the pseudolite transmitter is relatively stable, and the indoor environment has a high spatial resolution. For example, in the experiment, a pseudolite signal transmitter with six channels is selected, and each channel is connected with a transmitting antenna. Then, carrier phase difference refers to the difference of carrier phase between different antennas. Two antennas, respectively, transmit navigation signals to the receiver, and the carrier phase measurement equations can be expressed as:(1)$$\lambda N_{u}{}^{i}+\lambda \phi_{u}{}^{i}(t)=R_{u}{}^{i}+\delta t_{u}-\delta t^{i}+T_{u}{}^{i}+m_{u}{}^{i}+\varepsilon_{u}{}^{i}$$
(2)$$\lambda N_{u}{}^{j}+\lambda \phi_{u}{}^{j}(t)=R_{u}{}^{j}+\delta t_{u}-\delta t^{j}+T_{u}{}^{j}+m_{u}{}^{j}+\varepsilon_{u}{}^{j}$$

Among them, i and j represent the numbers of the two pseudolites, respectively; $R_{u}{}^{i}$ is the geometric distance from the pseudolite transmitter to the user receiver; $\delta t_{u}$ is the receiver clock deviation (relative system time); $\delta t^{i}$ is the pseudolite clock deviation (relative system time); $T_{u}{}^{i}$ is tropospheric delay; $m_{u}{}^{i}$ is multipath delay; $\varepsilon_{u}{}^{i}$ is the error term; λ is the carrier wave length; and N is integer ambiguity. The two equations are subtracted at the same time, resulting in:(3)$$\lambda N_{u}{}^{i,j}+\lambda \Delta \phi_{u}{}^{i,j}=\Delta R_{u}{}^{i,j}-\delta t^{i,j}+\Delta T_{u}{}^{i,j}+\Delta m_{u}{}^{i,j}+\Delta \varepsilon_{u}{}^{i,j}$$

The difference between the two equations at the same time is the carrier phase difference data used in this paper. Because of the same source, $\delta t^{i,j}$ and $\Delta T_{u}{}^{i,j}$ are eliminated, and multipath error $\Delta m_{u}{}^{i,j}$ and other errors $\Delta \varepsilon_{u}{}^{i,j}$ can be neglected because they are relatively small. Because of the fractional part of carrier phase, the integer ambiguity N is also eliminated. Furthermore, the carrier phase difference obtained by us is very stable. The stability of carrier phase difference of each group is tested by experiments. Therefore, this paper only uses carrier phase difference data between different antennas for indoor positioning. The following is validated by the measured data.

#### 2.2.1. Stability Test

In an indoor environment, we collect the carrier phase data of pseudolite at a certain location for a period of time to analyze its stability. The monitoring results of carrier phase difference of pseudolite are as follows.

In Figure 2, the carrier phase difference data of the pseudolite is relatively stable over a long period of time, with a fluctuation range of 0.02 to 0.03 cycle. It is worth noting that the cycle represents the oscillation period; in other words, the distance that the carrier propagates in one cycle is one wavelength. In reality, phase-locked loop (PLL) usually uses a carrier integrator to track the carrier phase change caused by Doppler shift. PLL must have errors in the measurement of the signal, which cannot be avoided. Usually the error comes from phase jitter. The accuracy of the measurement of the carrier phase by a general GPS receiver is 0.025 cycle. Therefore, the 0.02~0.03 cycle fluctuations we measured satisfy the stability requirements of fingerprint positioning, which provides a guarantee for high-precision positioning if in a quiet or relatively quiet environment.

#### 2.2.2. Spatial Resolution Test 

Theoretically, the higher the spatial resolution of the signal, the higher the positioning accuracy based on fingerprint matching. Therefore, we collected the carrier phase difference at four locations with an interval of 0.3 m in the indoor environment. The test results are as follows.

In Figure 3, the carrier phase difference has obvious resolution at four reference points 30 cm apart, which provides better resolution for fingerprint matching. It needs to be explained that the fluctuation of the blue curve is about 0.1 cycle, which is due to the existence of various noises in real measurement, which affects the measurement of carrier phase difference and reduces the positioning accuracy.

### 2.3. Overview of the System

This paper proposes an array pseudolite indoor positioning system. The system consists of multi-channel homogenous pseudolite base stations and data acquisition terminals. Two stages are needed to realize positioning: Offline fingerprint database construction phase and online positioning phase. The system diagram is shown in Figure 4.
During the offline phase. Firstly, the professional uses the positioning terminal to collect the carrier phase data of the array pseudolite antenna at the position of the known coordinates in the room, and then processes the data noise through the preprocessing method to obtain the carrier phase difference between the channels, and constructs the carrier phase fingerprint map. Then, the constructed data set is sent to the server for model training. Finally, the trained model is distributed to the positioning terminal.During the online phase. In the current indoor environment, users receive the pseudolite signal in real time, and realize the position prediction through the trained positioning model.

## 3. Proposed Method

In the research of indoor positioning technology, if the distribution of indoor signal sources in indoor space is modeled, the performance of an indoor positioning algorithm based on wireless signals will be greatly improved. However, the indoor environment is complex, and the signal transmission model is often difficult to determine. The ideal free-space transport model cannot be used in a real environment. Therefore, this paper proposes a deep learning network model to approximate the distribution characteristics of multi-channel signal carrier phase difference in an indoor environment, and then to realize the feature clustering under the latent space of fingerprint data. Finally, the clustered features are used as the input features of the location network model, and the regression network is used to predict the indoor location.

### 3.1. Variational Auto-Encoder (VAE)

VAE is an unsupervised deep generative model developed in recent years. The basic coding idea of VAE is based on the Gaussian mixture model (GMM). In simple terms, any kind of distribution can be decomposed into a superposition of several Gaussian distributions. At a certain location in the indoor environment, the collected observation x is composed of data of multiple pseudolite antennas, and the p(x) distribution can be regarded as an accumulation of the distribution of signals of different antennas in the Integral Domain. Unfortunately, computing p(x) is quite difficult, so the neural network is introduced to realize the encoding of the input data x, and the latent feature z is extracted, so that the originally disordered fingerprint data is clustered in the 2D latent space through the auto-encoder. The output of the VAE is the mean and variance of each dimension of data, so that the result of clustering is a continuous distribution. Each time through the randomly sampling from each latent state distribution to get a different vector as input, compared to the traditional auto-encoder (AE), VAE coding features are more diverse, and the model converges faster. In the decoding phase, the decoder learns not only the representation of a single point in the latent space, but the coding result of all points in the entire neighborhood. Moreover, it does not cause a large deviation between points in the nonlinear transformation process, thereby improving the fault tolerance of the decoder. The specific implementation process is as follows:

Firstly, the derivation process of the objective function of VAE is introduced. The goal of VAE is to learn parameter θ and maximize the marginal density function $p_{\theta }\left(x\right)=\int p_{\theta }\left(x|z\right)p_{\theta }\left(z\right)dz$. The conditional probability density $q_{\phi }\left(z|x\right)$ is usually introduced to approximate the posterior probability of the intractable $p_{\theta }\left(z|x\right)$, and there is also p(x,z)=p(x)p(z|x), so we use the joint distribution $q_{\phi }\left(x,z\right)$ to approximate $p_{\theta }\left(x,z\right)$, then according to the *KL* divergence [26] is chosen as measurement of the similarity between $q_{\phi }\left(x,z\right)$ and $p_{\theta }\left(x,z\right)$: (4)$$KL\left(p\left(x\right)\|q\left(x\right)\right)=\int p\left(x\right)\log\frac{p\left(x\right)}{q\left(x\right)}dx=E_{x∼p\left(x\right)}\left[\log\frac{p\left(x\right)}{q\left(x\right)}\right]$$

Therefore, the similarity of two joint probability distributions can be obtained from Equation (4):(5)$$KL\left(p_{\theta }\left(x,z\right)\|q_{\phi }\left(x,z\right)\right)=\iint p_{\theta }\left(x,z\right)\log\frac{p_{\theta }\left(x,z\right)}{q_{\phi }\left(x,z\right)}dzdx$$

*KL* divergence is the objective function. We hope that the two distributions are as close as possible. In the sampling process of the latent feature z, we can leverage a clever idea known as the reparameterization trick [26] which suggests that we randomly sample εfrom a unit Gaussian, and then shift the randomly sampledε by the latent distribution’s mean μ and scale it by the latent distribution’s variance σ. With this reparameterization, we can now optimize the parameters of the distribution while still maintaining the ability to randomly sample from that distribution. Finally, z can describe the characteristics of x, so have: (6)$$KL\left(p_{\theta }\left(x,z\right)\|q_{\phi }\left(x,z\right)\right)={Ε}_{x∼p_{\theta }\left(x\right)}\left[\int p_{\theta }\left(z|x\right)\log\frac{p_{\theta }\left(x\right)p_{\theta }\left(z|x\right)}{q_{\phi }\left(x,z\right)}dz\right]$$

Since there is $\log\frac{p_{\theta }\left(x\right)p_{\theta }\left(z|x\right)}{q_{\phi }\left(x,z\right)}=\log p_{\theta }\left(x\right)+\log\frac{p_{\theta }\left(z|x\right)}{q_{\phi }\left(x,z\right)}$, Formula (6) can be simplified as:(7)$$KL\left(p_{\theta }\left(x,z\right)\|q_{\phi }\left(x,z\right)\right)={Ε}_{x∼p_{\theta }\left(x\right)}\left[\log p_{\theta }\left(x\right)\right]+{Ε}_{x∼p_{\theta }\left(x\right)}\left[\int p_{\theta }\left(z|x\right)\log\frac{p_{\theta }\left(z|x\right)}{q_{\phi }\left(x,z\right)}dz\right]$$

There are, $KL\left(p_{\theta }\left(x,z\right)\|q_{\phi }\left(x,z\right)\right)\geq 0$ and $q_{\phi }\left(x,z\right)=q_{\phi }\left(x|z\right)q_{\phi }\left(z\right)$, so:(8)$$\log p_{\theta }\left(x\right)\geq -\left[\int p_{\theta }\left(z|x\right)\log\frac{p_{\theta }\left(z|x\right)}{q_{\phi }\left(x|z\right)q_{\phi }\left(z\right)}dz\right]$$
where $-\left[\int p_{\theta }\left(z|x\right)\log\frac{p_{\theta }\left(z|x\right)}{q_{\phi }\left(x|z\right)q_{\phi }\left(z\right)}dz\right]$ is the variation lower bound, the processing of the maximization likelihood function is transformed into the optimization of the EBO (Evidence Lower Bound Objective), which is further reduced to:(9)$$ELBO\left(\theta ,\phi ,x\right)=-KL\left(p_{\theta }\left(z|x\right)\|q_{\phi }\left(z\right)\right)+{Ε}_{z∼p_{\theta }\left(z|x\right)}\left[\log q_{\phi }\left(x|z\right)\right]$$

In Equation (9), in order to facilitate sampling, the hidden variable z is usually assumed to be z∼N(0,I); that is, a standard multivariate normal distribution.

The distribution $p_{\theta }\left(z|x\right)∼N\left(\mu \left(x\right),\sigma^{2}\left(x\right)\right)$ is fitted by constructing a simple neural network with input data x, mean μ(x), and variance $\sigma^{2}\left(x\right)$ as the output of the neural network. Finally, the generation model is based on two approaches discussed in the paper ‘Auto-Encoding Variational Bayes’ https://arxiv.org/abs/1312.6114, usually selected as multivariate Bernoulli distribution or Gaussian distribution.

### 3.2. Improved Model Structure

In the design idea of this paper, we hope to describe the distribution of the input fingerprint data x by means of the VAE-encoded latent variable z. In order to take advantage of the identification of location tags and the distribution of different signals at different locations, we designed a conditional VAE (CVAE) model. In the VAE codec training, the position label is used as a priori condition, and the signals at different positions in the room have their own unique distribution, which can reduce the influence of indoor signal fluctuation on the positioning result and play a role in adapting to the environment. At the time of model construction, the hidden layers of the encoder and decoder consist of four layers of convolutional layers, which can extract representative features. In the model training, we optimize the reconstruction loss, *KL* loss, and classification loss together. Finally, the latent variable z generated by the CVAE encoder is connected to the Softmax function to achieve positioning. At this time, $Softmax\left(y_{i};z\right)=\frac{e^{\left\{w_{i}z+b_{i}\right\}}}{\sum_{n}e^{\left\{w_{n}z+b_{n}\right\}}}$, where $w_{i}$ and $b_{i}$ represent the weight and bias of the i class, and z is the characteristic of the input data. In real-time positioning, the input data is encoded by the trained encoder, and the predicted position is obtained by DNN network regression. The network model constructed in this paper is shown in Figure 5.

Where x is carrier phase difference sequence, and y is the position coordination. Firstly, the two are simultaneously input into the encoder network of the pure convolutional layer [27], and the mean $\mu_{\theta }$ and variance $\sigma_{\theta }$ of x in the latent space distribution are obtained. The latent layer variable z is obtained by sampling according to the formula z=μ+σ⋅ε. At the time of decoding, the latent variable and the position label are input into the decoder to reconstruct the input data $\hat{X}$. Finally, the discriminator is used to optimize the loss and realize the training of the model. The specific implementation is as follows:

The variational lower bound of the conditional variation auto-encoder obtained by Equation (9) is:(10)$$ELBO\left(\theta ,\phi ,x,y\right)=-KL\left(p_{\theta }\left(z|x,y\right)\|q_{\phi }\left(z|y\right)\right)+{Ε}_{z∼p_{\theta }\left(z|x,y\right)}\left[\log q_{\phi }\left(x|z,y\right)\right]$$

Now, Monte Carlo log-likelihood estimation usually requires a large number of samples to be accurately estimated, or we use importance sampling to estimate conditional similarity. Then the formula $z∼p_{\theta }\left(z|x,y\right)$, $q_{\phi }\left(z|y\right)∼N\left(\mu_{y},Diag\left(\left\{\sigma_{y}^{2}\right\}\right)\right)$ is used to optimize the variational lower bound. It is worth noting that when calculating the model loss, we consider the reconstruction loss, *KL* loss, and classification loss together. The new variational lower bound is further obtained by p(x,z,y)=p(x|z)p(z|y)p(y) and Equation (7):(11)$$\begin{array}{cc} \log p_{\theta }\left(x\right) & \geq -KL\left(p_{\theta }\left(y|z\right)\|q_{\phi }\left(y\right)\right)+{Ε}_{z∼p_{\theta }\left(z|x\right)}\left[\log q_{\phi }\left(x|z\right)\right]-\sum_{y}p_{\theta }\left(y|z\right)\log\frac{p_{\theta }\left(z|x\right)}{q_{\phi }\left(z|y\right)}\\ & =ELBO\left(\theta ,\phi ,x,y\right) \end{array}$$
where $p_{\theta }\left(z|x\right)∼N\left(\mu \left(x\right),\sigma^{2}\left(x\right)\right)$, $q_{\phi }\left(x|z\right)∼N\left(\mu \left(z\right),1\right)$, $q_{\phi }\left(y\right)∼Categorical\left(y\right)$, and $p_{\theta }\left(y|z\right)$ can be obtained by fitting the Softmax function. Finally, we train a neural network as regression to learn the localization function mapping. The input data of the model is the carrier phase difference data with the position information, wherein the carrier phase difference is derived from the difference of the carrier phase between the antennas of the pseudolite, and the output is position coordinates in two-dimensional space. The details of the positioning algorithm are shown in the following Algorithms 1 and 2.

**Algorithm 1.** Model Training**Input**: Carrier Phase Fingerprint Data Set: $X=\left[x^{\left(1\right)},x^{\left(2\right)}\cdots x^{\left(n\right)}\right]\in R^{d\times A_{n}^{2}}$, d is the dimension of fingerprint data for each location, n is the number of location categories, $A_{n}^{2}$ refers to the number of combinations of pseudolite antennas. Location label: y**Output**: Representation: z and parameter: ϕ; θ, Classification model: $Model_{classifier}\left\{\left(z,y\right)\right\}$.1:  **Initialization Parameters:** Number of neurons for all layers;              The number of iterations (epochs);              Dimensions of latent Spaces;2:  **while**
{ϕ,θ} not converged **do**3:   D←getMinibatch()4:   $\mu_{\theta },\theta_{\theta }\leftarrow x,y$;5:   Sampling ε from N∼(0,I);6:   Sampling from the posterior $z\leftarrow q_{\phi }\left(z|x,y\right)$ using the flowing      Reparameterization trick: $z=\mu_{\theta }+\sigma_{\theta }\cdot \varepsilon$;  7:   Calculate the gradient of the variational lower bound L (Reconstruction loss and Classification loss and KL loss);8:   Minimize L;9:  **end while**10:  **while** Classification model Training **do**11:   Fit ∀{x,y}∈D train Classifier $Model_{classifier}\left\{\left(z,y\right)\right\}$12:  **end while**

**Algorithm 2.** Location Prediction**Input**: Real-time observation data: x and Classification model: $Model_{classifier}\left\{\left(z,y\right)\right\}$;**Output**: Predicted location y.1:  **for** each x
**do**2:    Predictive location by classifier: $y\leftarrow Model_{classifier}\left\{\left(z,y\right)\right\}$;3:  **end for**

## 4. Implementations and Evaluation

In this section, we verify the feasibility and positioning performance of the system through experiments. First, we compare the clustering effect of the proposed model with the common VAE model on the public MNIST data set. Next, in the test environment, the positioning performance of the positioning system is tested, including static positioning accuracy test and dynamic positioning accuracy test.

### 4.1. Experimental Methodology

As shown in Figure 6, we built the 7 × 10 m^2^ test environment in an office room and fix a single six-element array of pseudolite on the roof. The professional grids the indoor area at intervals of 30 cm and uses the total station to measure the position coordinates of each sampling point.

In the offline phase, pseudolite data of fixed time intervals is acquired at each sampling point using an intelligent terminal. Firstly, professionals use Ublox to analyze the quality and working status of indoor pseudolite data in real time, and also use the smart phone to obtain pseudolite data at the same location, and obtain the carrier phase value and the corresponding satellite number from the original observation data, as shown in the following figure. Then, the preprocessed data are sent to the server to construct a fingerprint database. It is worth noting that each location corresponds to data of multiple epochs when the fingerprint database is constructed. Finally, the deep neural network model is trained on the server side to extract data features.

In the online phase, the tester opens the positioning software, downloads the trained model from the server side, and receives the signal from the pseudo satellite antenna in real time in the test environment to realize position prediction. In Figure 7, the left side is the display result of the windows software U-CENTER v8.12, and the right side is the result obtained by using the navigation chip of the smartphone. The training data set is constructed by comparing and observing their stability.

### 4.2. Clustering Performance of the Model

In this paper, the MNIST data set (the MNIST data set comes from the National Institute of Standards and Technology. The training set consists of 250 different handwritten numbers, of which 50% are high school students and 50% are Census Bureau staff. The same proportion of handwritten digital data) is used to verify the clustering effect of the proposed network model and the commonly used variational auto-encoder in the latent space. The test results are shown in the figure below.

It can be seen from Figure 8, as the number of iterations increases, the chaotic input data achieves a better clustering effect in the implicit space. Compared with the ordinary VAE model, the proposed model has a better clustering effect, which can provide better feature knowledge for subsequent position prediction. At the same time, we also compared the clustering performance of our model with other commonly used models on the relevant data sets. The results are as follows:

Among them, MNIST, CIFAR10, REUTERS are integrated into Keras, which can be directly loaded for testing, BLE RSSI data set [29] and UJIIndoorLoc data set [30]. We use the deep learning framework Keras to implement the algorithm in Table 1. It can be seen from the test results that, compared with the mainstream clustering method, the variational auto-encoder with position information can achieve better clustering effect. The evaluation criteria of common clustering algorithms are that the spacing between similar classes is smaller and the spacing between different classes is larger. However, the clustering in the latent space in this paper makes some intersections of probability distributions of different classes by adding noise. The main reason for this is to avoid sampling in the blank feature area during the reparameterization trick, and also to reduce the data reconstruction error, and finally provide the power of knowledge for the positioning model.

### 4.3. Location Estimation Accuracy

The above sections verify the validity of the proposed model by examining the published data set. In this section, the performance of the positioning system will be evaluated from different states of static positioning and dynamic positioning.

#### 4.3.1. Verification of Positioning Accuracy in Static State

In order to verify the stability of the positioning system and the static positioning accuracy, the professional holds the smart terminal and stands still for a certain period of time at the reference point of the known coordinates. At the same time, in order to analyze whether the human body’s occlusion of the signal affects the positioning accuracy, the professional faces the different directions when performing the static positioning test. The test results are shown in the following figure.

In Figure 9, the blue circle is the position of the array pseudolite antenna, and the red circle is the reference point of the selected known position coordinates, where (A), (B), (C), (D) are the positioning results of the reference points 1, 2, 3, and 4, respectively. The horizontal and vertical coordinates are x and y in the Gaussian coordinate system, respectively, and the unit is meters. Among them, in order to clearly see the positioning results, this paper enlarges the local area of the positioning results, showing only ones, tenths, and percentiles in the horizontal and vertical coordinates. In order to describe the positioning result more intuitively, we calculate the positioning error value at different reference positions, as shown in the following figure:

It can be seen intuitively from Figure 10 that the positioning system proposed in this paper can achieve higher precision positioning performance in static positioning tests. The average positioning error is less than 10 cm, and the occlusion of pseudolite signals by the human body has little influence on the positioning accuracy.

#### 4.3.2. Verification of Positioning Accuracy in Walking State

In order to better evaluate the positioning performance of the proposed positioning system, we set different walking routes in the indoor environment as shown in Figure 11A–F. On the same walking route, the testers walk in a counterclockwise and clockwise order, and the test results are as follows:

In Figure 11, the red circle is the reference position point of the selected known coordinates, the blue point is the positioning result, and the red line is the real track. Among them, in order to clearly see the positioning results, this paper enlarges the local area of the positioning results, showing only two digits before and after decimal points in the horizontal and vertical coordinates. It can be seen from Figure 11A,B,D that although the testers pass the same reference point, the trajectories of the positioning have different deviations due to different walking directions. Therefore, it can be concluded that body occlusion has a certain impact on the positioning accuracy, but the impact is not large, and still meets the needs of indoor positioning. In the dynamic positioning error analysis, since the real position of the user cannot be obtained in real time, we select the reference points of some known coordinates on the motion track to evaluate the positioning accuracy, as shown in Table 2.

In order to compare different algorithms, the fingerprint database under the current experimental environment is constructed to collect pseudolite data at 30 cm intervals, including the arrangement and combination of carrier phases from six arrays. The k-Nearest Neighbor (KNN) and Support Vector Machine (SVM) models can be directly invoked through the deep learning framework Keras library. At the same time, the positioning algorithm in the reference [31,32] is also realized by using the framework. According to the set route, the experimenter walks different trajectories and compares the positioning accuracy. The experimental results are shown in Figure 12, Figure 13 and Table 3.

## 5. Conclusions and Future Work

In this paper, an indoor positioning system using array pseudolite is proposed. Aiming at the high spatial resolution of pseudolite carrier phase difference and high signal stability, a fingerprint matching positioning method based on carrier phase difference is designed. In the model construction stage, the VAE deep learning network model with position information was designed by using the deep learning framework Keras, which was formed by stacking pure four-layer CNN networks. By testing the model, it has a better clustering effect in the latent space than the traditional VAE, and can extract more representative features. In the real indoor office environment, this paper verifies the positioning performance of the positioning system through a large number of experiments. The test results show that the average positioning error of the positioning system is less than 10 cm in the static state. In the motion state, this paper compares several commonly used localization algorithms based on fingerprint database matching. The average positioning accuracy of our positioning system is 0.39 m, and 95% of the positioning error is less than 0.85 m, which has better positioning performance.

In the future, our work will focus on the use of semi-supervised learning to achieve a complete fingerprint database using a small number of indoor location labels, thereby improving the efficiency of fingerprint database construction and facilitating the application.

## Figures and Tables

**Figure 1 sensors-19-04420-f001:**
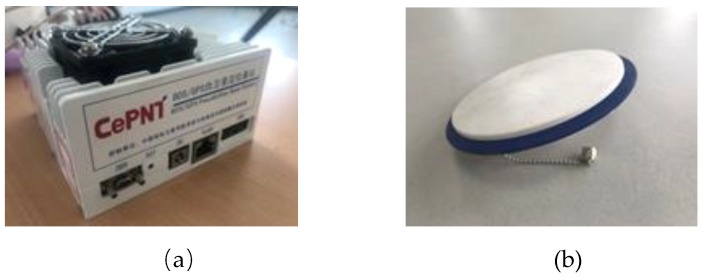
(**a**) pseudolite base station; (**b**) pseudolite antenna.

**Figure 2 sensors-19-04420-f002:**
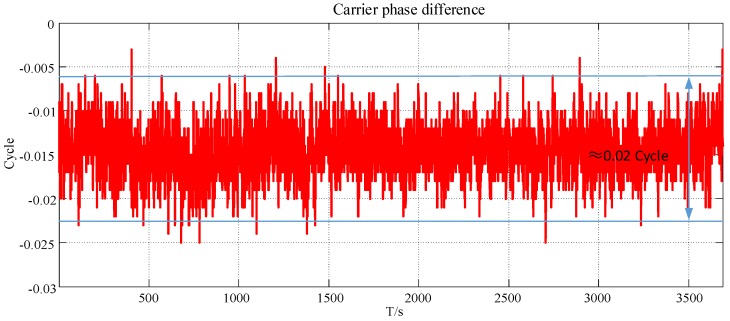
Observation results of carrier phase difference stability.

**Figure 3 sensors-19-04420-f003:**
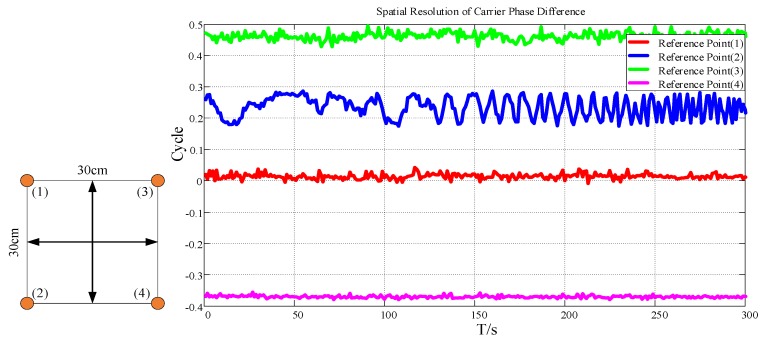
Spatial resolution of carrier phase difference.

**Figure 4 sensors-19-04420-f004:**
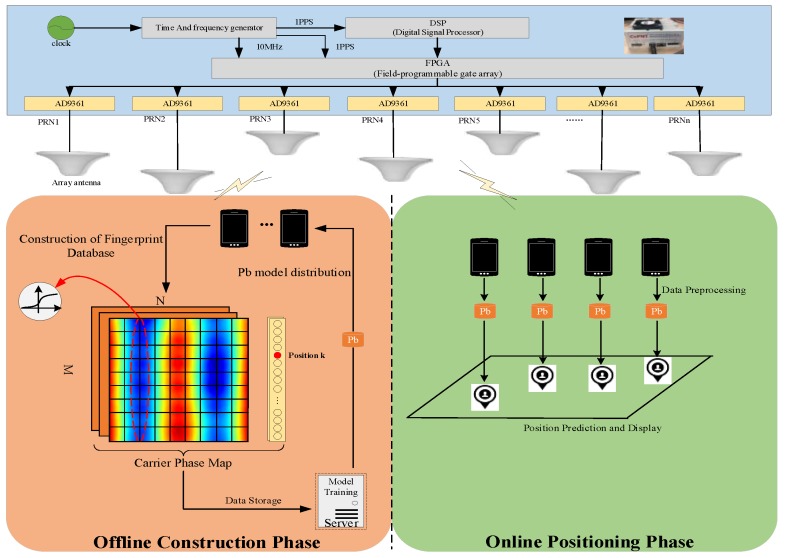
Array pseudolite positioning system architecture diagram.

**Figure 5 sensors-19-04420-f005:**
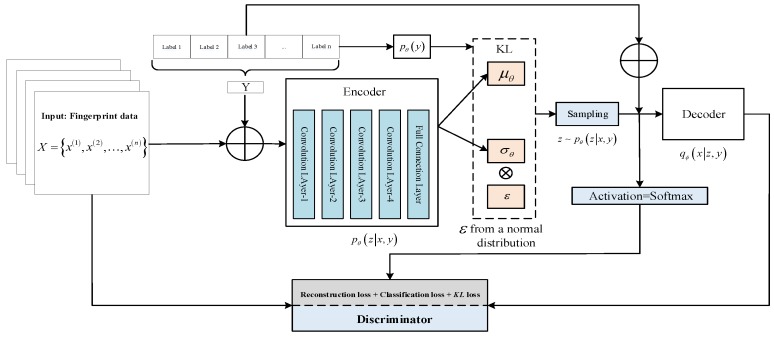
Structure diagram of conditional variation auto-encoder network.

**Figure 6 sensors-19-04420-f006:**
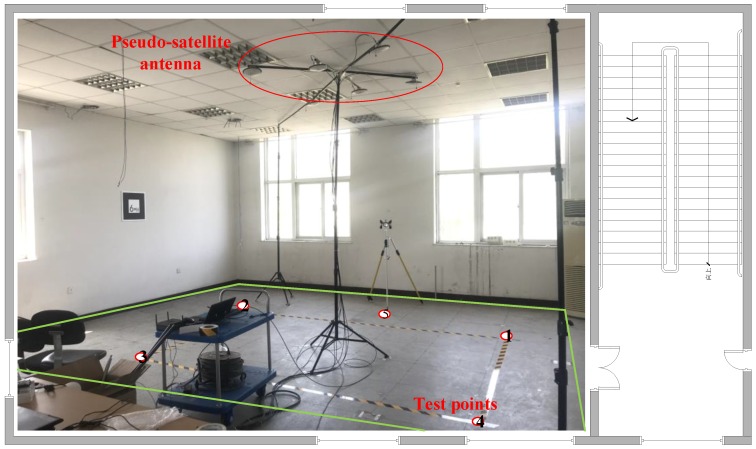
Diagram of the test environment.

**Figure 7 sensors-19-04420-f007:**
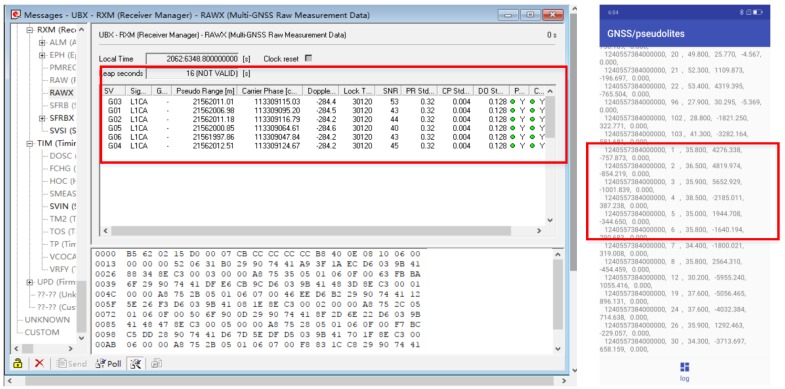
Ublox signal acquisition software and smartphone acquisition software.

**Figure 8 sensors-19-04420-f008:**
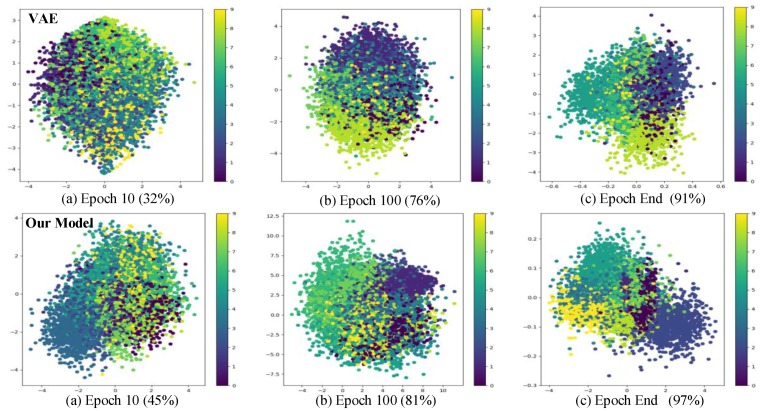
The clustering effect of our model in 2D latent space on MINIST data set.

**Figure 9 sensors-19-04420-f009:**
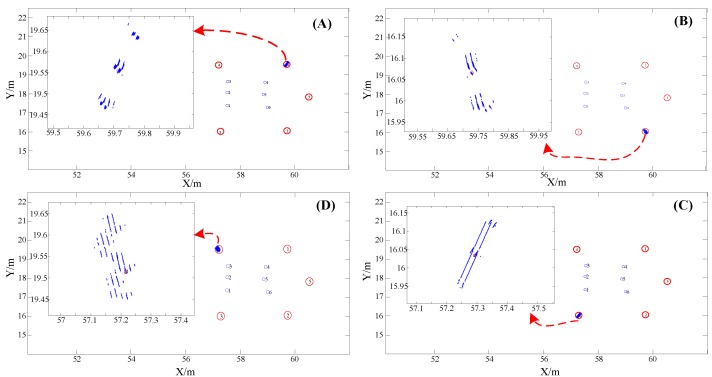
Test results for static positioning at different locations.

**Figure 10 sensors-19-04420-f010:**
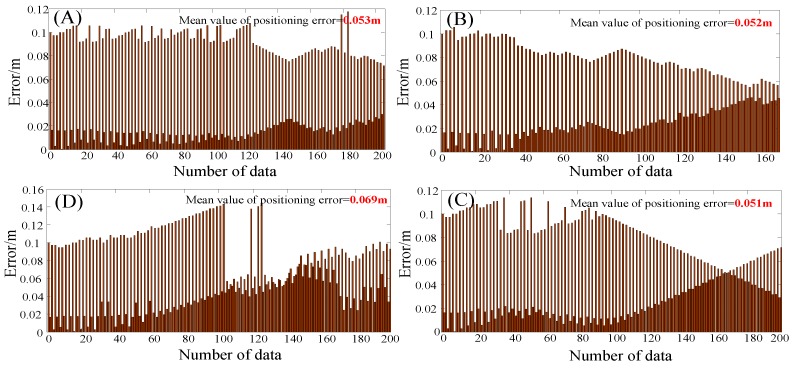
Error results of static positioning at different positions.

**Figure 11 sensors-19-04420-f011:**
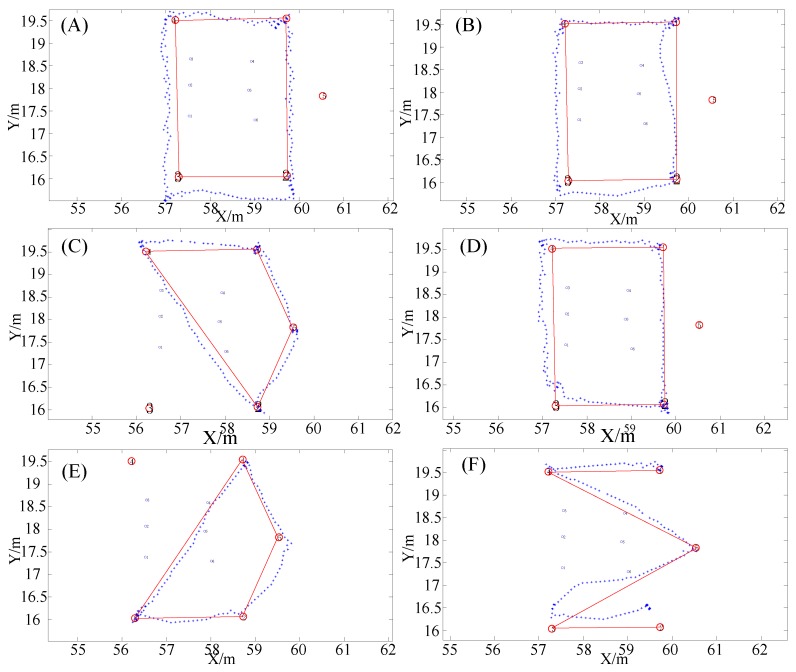
Positioning test results under walking conditions with different trajectories.

**Figure 12 sensors-19-04420-f012:**
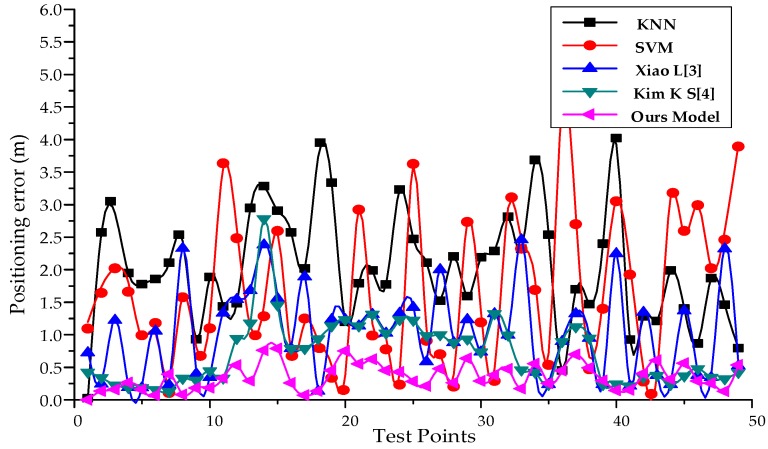
Comparison of positioning errors of different positioning algorithms.

**Figure 13 sensors-19-04420-f013:**
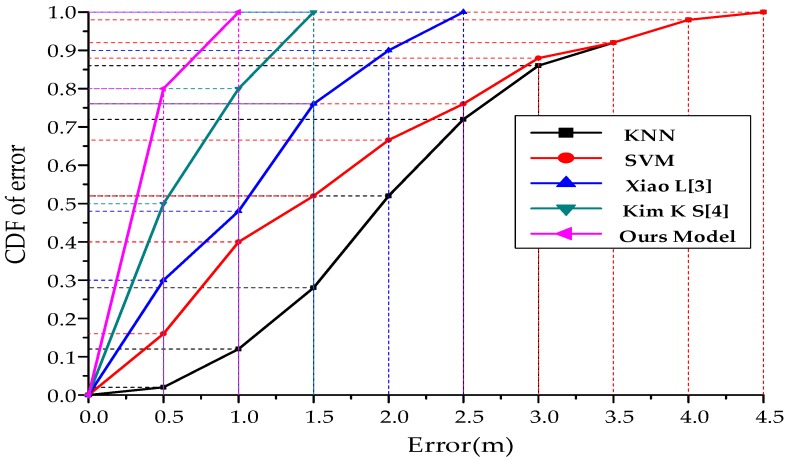
Cumulative distribution function of errors from different positioning algorithms.

**Table 1 sensors-19-04420-t001:** Comparisons of clustering performance of various models on different data sets.

Method	MNIST	CIFAR10	REUTERS	BLE RSSI	UJIndoorLoc
GMM	55.37 (±0.08)	55.46 (±0.10)	56.01 (±0.11)	45.56 (±0.12)	50.23 (±0.12)
AE + GMM	84.56 (±0.11)	73.59 (±0.08)	71.18 (±0.11)	74.61 (±0.11)	85.91 (±0.11)
VaDE [28]	94.46 (±0.1)	88.36 (±0.05)	79.58 (±0.10)	89.34 (±0.04)	91.85 (±0.04)
Our Model	97.77 (±0.08)	90.11 (±0.04)	82.07 (±0.08)	91.36 (±0.05)	92.96 (±0.02)

**Table 2 sensors-19-04420-t002:** Comparison of location accuracy in motion state.

Path	True Position		Measured Position		Error of Position	
	X (m)	Y (m)	X (m)	Y (m)	X (m)	Y (m)
(A) 1–2–3–4–1	4,235,057.22	530,519.52	4,235,057.11	530,519.79	0.11	0.27
	4,235,057.29	530,516.03	4,235,057.77	530,515.61	0.48	0.32
	4,235,059.73	530,516.07	4,235,059.24	530,515.49	0.49	0.58
(B) 1–4–3–2–1	4,235,057.22	530,519.83	4,235,057.29	530,519.93	0.07	0.10
	4,235,057.29	530,516.03	4,235,057.48	530,516.15	0.17	0.12
	4,235,059.73	530,516.07	4,235,059.68	530,516.11	0.05	0.04
(C) 1–5–2–4–1	4,235,059.71	530,519.55	4,235,059.63	530,519.67	0.08	0.12
	4,235,060.53	530,517.83	4,235,060.62	530,517.79	0.09	0.04
	4,235,057.22	530,519.52	4,235,057.52	530,519.71	0.30	0.19
(D) 2–1–4–3–2	4,235,059.71	530,519.55	4,235,059.77	530,519.48	0.06	0.07
	4,235,057.22	530,519.52	4,235,057.34	530,519.44	0.12	0.08
	4,235,057.29	530,516.03	4,235,057.34	530,516.18	0.05	0.15
(E) 3–1–5–2–3	4,235,057.29	530,516.03	4,235,057.34	530,516.12	0.05	0.09
	4,235,060.53	530,517.83	4,235,060.65	530,517.75	0.12	0.08
	4,235,059.73	530,516.07	4,235,059.51	530,516.11	0.22	0.04
(F) 1–4–5–3–2	4,235,059.71	530,519.55	4,235,059.76	530,519.64	0.05	0.09
	4,235,057.29	530,516.03	4,235,057.27	530,516.26	0.02	0.23
	4,235,059.73	530,516.07	4,235,059.49	530,516.49	0.24	0.42

**Table 3 sensors-19-04420-t003:** Comparison of location accuracy (m) of common algorithms based on fingerprint matching.

Algorithm	KNN	SVM	Xiao L [6]	Kim K S [7]	Our Model
Mean Error(m)	2.03	1.56	1.03	0.79	0.39
95%Error(m)	3.77	3.76	2.25	1.43	0.85

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
