# Peer review of "An Innovative Fingerprint Location Algorithm for Indoor Positioning Based on Array Pseudolite"

_sensors, 2019, doi:10.3390/s19204420_

Round 1

Reviewer 1 Report

Comments :

Page 1 :

In your abstract, you should specify: “Variational auto-encoding (VAE)”, since you use the acronym later in the paper.

Page 2 :

“Wi-Fi fingerprint positioning [6,7], Radio Frequency Identification (RFID) [8], sound wave [9], visible light [10], Ultra-Wide Band (UWB) [11], BluetoothLow Power (BLE) [12,13], infrared [14], motion sensor based inertial navigation [15,16] , etc.”

You enumerate the different methods. You should explain in a few words the principle of each one. This would show the originality of your own.

“The near far effect” is the main reason why pseudolites are pulsed. It is interesting if you have a network of pseudolites. Here you have 6 signals, but they are so spatially close that the near far effect is probably weak.

Page 3:

In preliminary section 2, you should add some equations to show exactly what you are measuring and what will be use in your algorithm. For now, this is not clear. You are talking about phase differences but between what and what? Signal from two different antennas or the same antenna with GPS and Beidou. This is unclear, I guess it is the first solution and I should not have to guess. It must be written explicitly.  Machine learning is interesting, but the data collected to make it work is an essential point.

In the case of the difference between pseudolite, with 6 antennas, we have 30 differences possible (60 is we consider Beidou and GPS). Did you use all of them in your fingerprint database?

“multi-channel signal transmitter”  and “array antenna”: one source with several channel (6 in your experimentation) that are wiredly linked to antennas that broadcast the signals. Is my description correct?

“By using different pseudo-range of different array channels, the clock deviation between receiver and pseudo-range is eliminated, and the time synchronization problem of traditional pseudolite is avoided.”

The reason why the synchronization issue is avoided here is because all the signal broadcast by the pseudolite have the same clock. The difference eliminates this common clock. Here you see the interest of equations: the answer is direct.

Page 4:

“the carrier phase difference between channels is relatively stable” : My guess was correct, but you should have specified this before.

Figure 2: what is the significance of the “≈0.02 cycle” in the middle of the graph?

“which provides a guarantee for high precision positioning” : if you are static and with a quiet environment around.

Figure 3: The blue curve (ref point 2), seems to be more unstable with a kind of oscillation whose amplitude (0.1 cycle) is not negligible. Do you have an explanation? This is in contradiction with what you say about stability.

“the carrier phase difference between the same two antennas”: this is a clearer formulation, you carry out the difference between two signals from two distinct antennas.

 Page 9:

The number of antennas should be justified. Does 6 is a minimum? What happens with 5, 4, etc.?

Page 10:

Figure 7 is not easy to read.

Is there a reason to choose 50% from students and 50 % from Census bureau staff? Do you observe any difference?

Page 11:

Figure 9 and others. The scales on axis x and y are not appropriate. You express them in hundred thousands of meters, so only the last number is significant. Center your scale to have numbers in meters. The Gaussian scale is not relevant in these figures.

Page 12:

“the human body has little influence on the positioning accuracy”: the body of the holder of the smartphone, but what about the others around?

General comments:

One point that is original with machine learning method is that geometry is less important than for classical method. So you do not have the classical problematic of the dilution of precision.

It would be interesting to know the range of the method, i.e. the maximum size of the room that is covered by one pseudolites. Indeed, the displacement you carry out is in a area of approximately 2.5 m by 3 m. I ask the question since it appears to me that a larger room could lead to a redundancy of the phase difference values (the same phase difference could be associated to several positions).

Have you try to carry out the same method with pseudorange difference? Of course the sensitivity to multipath probably leads to very inaccurate differences, but this is the principle of fingerprint: the multipath are here, but the cartography takes them into account. Maybe this variation is changing according to the changes in the environment, but the carrier phase difference is submitted to the same issue.

There is a strong hypothesis that justifies your approach but which is not so obvious: the error due to multipath and occlusion can be model with a machine learning procedure. I would say that it is true if the environment is not changing too quickly or has always the same “density of change”. Anyway, does the model still match if you have ten peoples walking around one day and two peoples another day?

A point that you do not address in your paper: the pseudolite regulation. You mention the interest of pseudolites in terms of hardware. However, today pseudolite is not forbidden but strongly limited, in terms of transmission power and deployment, in most countries. Could you add some words about this question (power requirement, etc.)?

Author Response

Thank you very much.

The paper has been thoroughly revised.

Reviewer 2 Report

The manuscript proposed an innovative fingerprint location algorithm for indoor positioning based on array pseudolites. The major and minor problems of the manuscript are:

Major:

The authors said “the system transmits a unique C/A code signal compatible with GPS/BDS satellite signal, and data acquisition and location are realized by using smart phones.” But, if it is an unique signal, how can it be process by common smart phone? I do not think the ephemeris of the system is same as GPS/BDS. There are two stages in the algorithm: clustering and location prediction. The authors need to discuss why two steps are needed? Can we just use location prediction rather than clustering? For the clustering method based on VAE, what is training? Carrier phase data with known position? Are the authors mean the fraction part of carrier pseudo-range? Please make it clear what is measured in receiver and what is the output (not just say the input data of the model is training data set and the output is position coordinates in two-dimensional space). Also in VAE, what is the space of grid in position during the training? If it is 30cm, how can we have an accuracy at cm level? And the authors should compared the proposed methods with traditional KNN. Commonly, a good KNN methods can also have a 90% level accuracy.

Minor:

What is the label means in figure 5? Is it an ID or a coordination? What is Cn in table of algorithm 1? The results shown in figure 9 has special patterns. Can the authors give more comments on it?

Author Response

(The authors gave the same response as above.)
